# The History of Carbon Monoxide Intoxication

**DOI:** 10.3390/medicina57050400

**Published:** 2021-04-21

**Authors:** Ioannis-Fivos Megas, Justus P. Beier, Gerrit Grieb

**Affiliations:** 1Department of Plastic Surgery and Hand Surgery, Gemeinschaftskrankenhaus Havelhoehe, Kladower Damm 221, 14089 Berlin, Germany; fivos.megas@gmail.com; 2Burn Center, Department of Plastic Surgery and Hand Surgery, University Hospital RWTH Aachen, Pauwelsstrasse 30, 52074 Aachen, Germany; jbeier@ukaachen.de

**Keywords:** carbon monoxide, CO intoxication, COHb, inhalation injury

## Abstract

Intoxication with carbon monoxide in organisms needing oxygen has probably existed on Earth as long as fire and its smoke. What was observed in antiquity and the Middle Ages, and usually ended fatally, was first successfully treated in the last century. Since then, diagnostics and treatments have undergone exciting developments, in particular specific treatments such as hyperbaric oxygen therapy. In this review, different historic aspects of the etiology, diagnosis and treatment of carbon monoxide intoxication are described and discussed.

## 1. Introduction and Overview

Intoxication with carbon monoxide in organisms needing oxygen for survival has probably existed on Earth as long as fire and its smoke. Whenever the respiratory tract of living beings comes into contact with the smoke from a flame, CO intoxication and/or inhalation injury may take place. Although the therapeutic potential of carbon monoxide has also been increasingly studied in recent history [1], the toxic effects historically dominate a much longer period of time.

As a colorless, odorless and tasteless gas, CO is produced by the incomplete combustion of hydrocarbons and poses an invisible danger. CO enters the human body through the inhalation of flue gases and can cause tissue hypoxia due to its affinity for the hemoglobin molecule, i.e., about 240 times higher than that of oxygen [2]. The oxygen is replaced and carboxyhemoglobin (COHb) is formed [3]. Organs that have a high oxygen demand and thus depend on high blood flow can be severely affected [4]. For example, due to its affinity for myoglobin, which is also 60 times greater than that of oxygen, the replacement of oxygen leads to cardiac depression and hypotension [5]. The exact pathophysiological mechanism is not yet fully understood. However, the toxic effect is attributed to the binding of CO to cytochrome oxidase and the inhibition of the electron transport chain [6].

Intoxication with CO causes thousands of deaths each year, as shown by a large number of the most recent studies published to date [7,8,9]. A brief overview of symptoms caused by CO poisoning includes nausea (40%), headache (46%), dyspnea (20%) and tachycardia (41%). Many patients also complain of dizziness and vomiting [9,10].

The diagnosis of CO intoxication involves several parameters. The basis is the measurement of the percentage of COHb, which can be detected by arterial blood gas examination. Alternatively, non-invasive CO-oximetry can be performed, especially in a preclinical setting [11]. Using these techniques, highly elevated values can be measured that differ greatly from the normal ratio of carboxyhemoglobin to hemoglobin of 2–3% in non-smokers and 5-9% in smokers [12]. However, this basic diagnosis must be supported, especially in severe cases, by further investigations such as thorough clinical examination, different laboratory markers and radiological diagnostics [13,14,15].

## 2. Literature Research

The literature search for this review was conducted with PubMed and Google Scholar using the following keywords: “carbon monoxide intoxication history”, “inhalation injury history” and “burns history”. In addition, we cross-checked reference lists from eligible publications and relevant review articles to identify additional studies. Our search in the relevant literature revealed 38 PubMed entries up to the year 1945, about 1850 entries from 1946 to 1975, about 2035 entries from 1976 to 2000 and the current literature from 2001 to 2020 has about 2564 entries. A time bar of the available publications on PubMed is shown in Figure 1.

Further, the search in PubMed revealed that there are some publications concerning the reproduction of burn medicine in antiquity [16,17] but none dealing specifically with CO intoxication. This, however, could be observed for the first time in the early 20th century [18].

## 3. 1900 to 1945

A total of 38 PubMed entries dealing with CO intoxications could be found in the period up to 1945. The first published article from 1906 involved animal experiments. The authors Nasmith and Graham investigated the hematology of carbon monoxide poisoning [18]. The methodology of their experiments is described in the following paragraph:


*Twelve guinea pigs were taken, six males and six females, and placed in two cages in the gas chamber after making careful estimations of the white and red blood corpuscles and haemoglobin. Gas was then allowed to mix with the air drawn through the chamber until the mixture was of such a strength that 250/0 of the haemoglobin of the guinea-pigs was saturated with carbon monoxide.*


The authors discovered that carbon monoxide prevents the normal supply of oxygen to the tissue and thus disrupts the metabolism of cells. They concluded a massive toxic effect of CO. The guinea pigs that lived constantly in a diluted carbon monoxide atmosphere during this experiment showed a reduced oxygen transport capacity of the blood. However, their organisms were able to compensate for the lack of oxygen by increasing the hemoglobin concentration and the number of erythrocytes. Furthermore, they found that the effect of carbon monoxide on increasing the number of erythrocytes was similar to the effect of high altitudes [18].

Another article from this period described the problems of military mining in cases when explosive charges do not detonate completely and continue to burn. Thus, very highly concentrated CO vapors were produced, which could have devastating consequences in a narrow mine [19]. The symptoms of CO intoxication were described for the first time in humans. The medical term used here for the clinical picture is “gassing”. In addition, this article also describes the first therapeutic approaches for CO intoxication [19].

*Treatment is summed up in warmth, oxygen, artificial respiration, rest. Circulatory stimulants, such as hot coffee, are valuable. Strychnine has been found useful for stimulating the respiratory centre, but chief reliance is to be placed upon artificial respiration by Schäfer’s method* [20], *combined with the administration of oxygen* [19].

Of course, some of these therapeutic approaches are outdated nowadays, but the central aspect of the therapy of CO intoxication was already recognized at that time, namely the administration of highly concentrated oxygen [19].

## 4. 1945 to 1980

During this time, due to industrial innovations such as the switch from coal to natural gas for the domestic supply, the number of CO intoxications began to decrease [21]. Apart from this aspect, this may also have been due to a growing number of alternative methods to commit suicide, in particular by using tranquilizers and antidepressants developed at that time [21]. However, cases of CO intoxication were still associated with high morbidity and mortality, resulting in more research into therapeutic options during this period. One of these new therapeutic methods was hyperbaric oxygen therapy (HBO), which will be discussed in a separate section due to its great importance [22,23]. Diagnostics also expanded in the 1970s, with the first devices, such as the Dräger test tube, that could accurately measure CO concentrations in exhaled breath [24]. This was certainly a significant innovation pre-clinically and in emergency situations. The initial clinical algorithm in 1970 is very similar to that used today.

*In selecting a form of therapy there are two aspects to consider: firstly, the prevention of death and, secondly, the reduction of neuropsychiatric sequelae such as those described by H. Garland and J. Pearce. Carboxyhaemoglobin should be eliminated as quickly as possible because its presence alters the dissociation curve of the remaining oxyhaemoglobin, impeding oxygen release to the tissues* [21].

Furthermore, it was first observed that long-term psychiatric sequelae could occur after CO intoxication. A follow-up study of 74 patients by Smith et al. in 1973 showed that the state of consciousness upon admission to hospital in the acute phase of poisoning correlated significantly with the development of severe neuropsychiatric sequelae. These findings underpin the importance of prompt and effective treatment of carbon monoxide poisoning and the need to follow up all clinical cases in anticipation of a recurrent course or the development of sequelae [25].

Nevertheless, Barret et al. indicated that, during this time, CO intoxications were often overlooked [2]. They conducted a trial in Grenoble, France, to determine the true incidence of missed or misdiagnosed CO intoxications. Misdiagnoses were unusually high, resulting in the launch of a public information campaign. The initial high rate of misdiagnosis (30% in 1975–1977) declined after the campaign (12% in 1978 and 5% in 1980), although the rate of hospitalization for confirmed CO poisoning increased substantially [2].

## 5. From 1980 to Present

Today, it is known that immediate intervention in CO intoxication is crucial. Triage and transfer to specialized hospitals as soon as possible after stabilization of the patient’s cardiopulmonary status are essential. For this purpose, vital parameters such as heart rate, blood pressure and percutaneous oxygen/CO-saturation are continuously monitored. In case of respiratory insufficiency, immediate intubation and ventilation with 100% O_2_ are required until further therapy can be initiated in an intensive care unit [9,12]. As described above, the administration of highly concentrated oxygen in the prehospital setting is essential in cases of suspected CO intoxication [26,27]. The Glasgow Coma Scale (GCS) is the most suitable way to immediately assess the neurological status of the patient [9,28]. These algorithms have existed since and even before the 1990s and have not changed much since then [26].

Innovations in recent years that have contributed significantly to the improvement of therapeutic options for patients with CO intoxication and inhalation trauma include the improvement of ventilation of intensive care patients. For example, it was shown that the use of lower tidal volumes (TVs) when ventilating patients with acute lung injury and acute respiratory distress syndrome reduces mortality in these patients by 22% [29]. The corresponding study was discontinued, as mortality in the group treated with lower tidal volumes was highly significantly lower than that of the group treated with high tidal volumes [29]. New standards like this were also adopted for the treatment of CO intoxication.

Furthermore, the long-term effects, such as inflammatory changes in the airway, are nowadays closely scrutinized in order to further reduce morbidity and mortality after CO intoxication [30].

## 6. History of Hyperbaric Oxygen Therapy for CO Intoxication

The first hyperbaric oxygen (HBO) chamber applied for medical purposes was described in 1622 [31]. In the 19th century, HBO chambers were used for the therapy of diseases such as tuberculosis, anemia and cholera [31]. Today, HBO chambers are also very promising in experimental settings, such as, e.g., conditioning cells for tissue engineering [32]. However, in the late 19th century, the first application of an HBO chamber was used in connection with CO intoxication. In 1895, Haldane could not poison a mouse in a chamber with a high oxygen concentration and a low CO concentration [33].

*The chief aim of the present investigation has been to determine experimentally the causes of the symptoms produced in man by carbonicoxide, and particularly the relation of the changes in the blood to the symptoms, to the percentage of carbonic oxide breathed, and to the period during which the inhalation is continued* [33].

It took several years for the idea of using HBO in CO intoxication to be revived. Smith et al. and Churchill-Davidson et al. first described the benefits of HBO therapy in CO intoxicated patients [34]. Since then, HBO therapy has remained a controversial topic and a definite recommendation cannot be made, as Buckley et al. show in their systematic review [35]. The first major studies concerning this topic were performed in the 1980s. A clinical trial by Raphael et al. compared the use of normobaric oxygen (NBO) with HBO. The results of this study showed that HBO therapy was not beneficial in patients who did not lose consciousness during carbon monoxide intoxication, regardless of their carboxyhemoglobin level. In this study, neither a positive effect nor a negative effect of HBO therapy could be demonstrated [36].

Further large randomized trials followed in the 1990s. Scheinkestel et al. showed that, in comparison with NBO, HBO therapy had no benefits and may have even worsened the outcome and was therefore not recommended by the authors [37]. Another randomized controlled trial with 179 patients by Annane et al. in 2011 also showed critical results regarding HBO therapy. No evidence of superiority of HBO over NBO in patients with a transient loss of consciousness could be shown. Furthermore, a second HBO session in comatose patients was associated with a worse outcome [38].

In contrast to these results, other studies have shown that a large majority of patients benefit from hyperbaric oxygen therapy by effectively minimizing late neurological sequelae as well as the development of brain edema and pathological changes to the central nervous system [39,40,41]. In 2002, Weaver et al. published a double-blind, randomized trial in the New England Journal of Medicine. In this study, NBO was compared with HBO therapy with respect to cognitive sequelae after acute carbon monoxide poisoning. The results showed that, six weeks after CO poisoning, neurological pathologies detected by neuropsychological testing were significantly less frequent in the HBO group (25 vs. 46.1%; *p* = 0.007) [42].

Currently, there are only a few national and no international guidelines on how to treat CO intoxication. For example, according to the “Clinical Guidance for Carbon Monoxide (CO) Poisoning”, as published by the U.S. CDC (The Centers for Disease Control and Prevention), HBO therapy shall “*be considered when the patient has a COHgb level of more than 25–30%, there is evidence of cardiac involvement, severe acidosis, transient or prolonged unconsciousness, neurological impairment, abnormal neuropsychiatric testing, or the patient is ≥36 years in age. HBO is also administered at lower COHgb (<25%) levels if suggested by clinical condition and/history of exposure*” [43].

According to the recommendations of the U.K. Department of Health and the NHS England, a “*COHb concentration of >20% should be an indication to consider hyperbaric oxygen and the decision should be taken on the basis of specific indications, i.e., loss of consciousness at any stage, neurological signs or symptoms other than headache, myocardial ischaemia/arrhythmia diagnosed by ECG, or pregnancy*” [44,45].

In Germany, a national guideline is currently under development under the auspices of the DIVI (German Interdisciplinary Association for Intensive Care and Emergency Medicine) which shall be finished and published soon (May 2021) [46].

In summary, a common opinion nowadays seems to be that routine HBO treatment cannot be recommended in general, but may be beneficial in patients with severe intoxication [35].

## 7. Conclusions

CO intoxication can occur in all living beings and has been a problem since fire and its vapors have existed on Earth. The etiology, diagnosis and specific treatment for CO intoxications have been studied very carefully over the years. Recovery, healing and therapeutic strategies of affected patients have been developed only in the 20th century and have been refined since then. As the controversial debate of HBO therapy shows, the destination of this journey has not been reached even today.

## Figures and Tables

**Figure 1 medicina-57-00400-f001:**
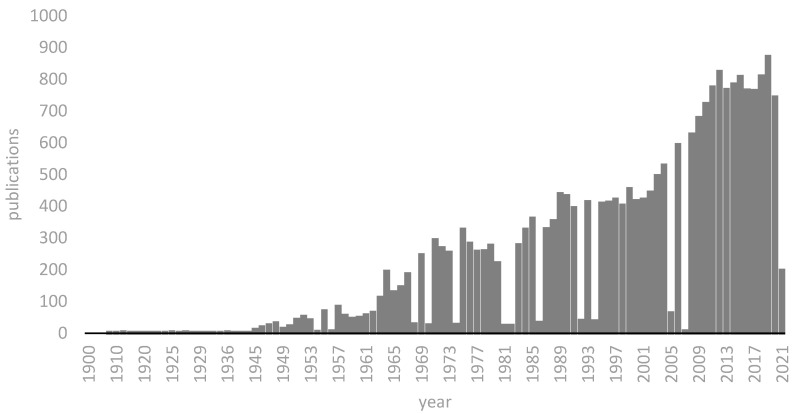
A time bar of the publications on Co intoxication available on PubMed across time.

## Data Availability

Not applicable.

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
