# Peer review of "The History of Carbon Monoxide Intoxication"

_medicina, 2021, doi:10.3390/medicina57050400_

Round 1

Reviewer 1 Report

This study is not different from the previous version, which was withdrawn. Nevertheless, it is an appropriate paper to demonstrate the significant steps of the knowledge and treatment of carbon monoxide poisoning. 

Literature is adequate as well as the style

Author Response

Thank your for the positive feedback.

Reviewer 2 Report

Well organized content and originality

Author Response

Thank your for the positive feedback.

Reviewer 3 Report

In general, very well done!  Just a few particular comments,...

  • It would be interesting to make a figure of the number of CO publications across time with a bar graph.
  • Some editing for English should be done including replacing subjunctive with indicative mood where indicated. 

Author Response

Thank you for your very positive impression and the constructive minor aspects.
As suggested, we added a figure demonstrating an overview of the publications of CO-intoxication over the different years. We appreciate this great idea.
Your further suggestions were to format the language style, e.g. changing conjunctive to indicative. We crosschecked the manuscript again with the help of a native speaker and were not able to find these problems of language style. However, if these problems still should be very dominant, we would be very happy to know the exact page and line that should be corrected.